# Antiproliferative Effect and Mediation of Apoptosis in Human Hepatoma HepG2 Cells Induced by Djulis Husk and Its Bioactive Compounds

**DOI:** 10.3390/foods9101514

**Published:** 2020-10-21

**Authors:** Dom-Gene Tu, Charng-Cherng Chyau, Shih-Ying Chen, Heuy-Ling Chu, Shu-Chen Wang, Pin-Der Duh

**Affiliations:** 1Department of Nuclear Medicine, Ditmanson Medical Foundation, Chia-Yi Christian Hospital, Chia-Yi 600, Taiwan; 03587@cych.org.tw; 2Department of Biomedical Sciences, National Chung Cheng University, Minhsiung, Chia-Yi 62102, Taiwan; 3Research Institute of Biotechnology, Hungkuang University, Taichung 43302, Taiwan; ccchyau@msa.hinet.net; 4Department of Health and Nutrition, Chia Nan University of Pharmacy and Science, Tainan 71710, Taiwan; shihying@mail.cnu.edu.tw; 5Department of Food Science and Technology, Chia Nan University of Pharmacy and Science, Tainan 71710, Taiwan; chuheuy@mail.cnu.edu.tw (H.-L.C.); shujen@mail.cnu.edu.tw (S.-C.W.)

**Keywords:** antiproliferation, apoptosis, djulis husk, quercetin, kaempferol

## Abstract

The antiproliferative effect and mediation of apoptosis in human hepatoma HepG2 cells induced by djulis husk and its bioactive compounds was investigated. The ethanolic extracts of djulis husk (EEDH) at 50, 250, and 500 µg/mL induced remarkable cytotoxicity on HepG2 cells. By flow cytometry analysis, EEDH slowed down the cell cycle at the Sub-G0 phase after 24 h of incubation. Moreover, all EEDH treatment induced an apoptotic response in HepG2 cells. EEDH-induced apoptosis was associated with the attenuation of mitochondrial transmembrane potentials (ΔΨ_m_), an increase in Bax/Bcl-2 ratio, activation of caspase-3, and poly(ADP-ribose)polymerase (PARP) cleavage, as well as an increase in reactive oxygen species (ROS) generation. According to the HPLC-DAD and HPLC-MS/MS analysis, quercetin and kaempferol derivatives and another sixteen compounds were present in EEDH. Quercetin and kaempferol at 25–150 μM showed antiproliferative action and induced apoptosis on HepG2 cells, which may in part account for the anticancer activity of EEDH. Overall, EEDH may be a potent chemopreventive agent due to apoptosis in HepG2 cells.

## 1. Introduction

A World Health Organization (WHO) survey indicated that cancer is the second leading cause of death globally and accounted for an estimated 9.6 million deaths in 2018. Liver cancer is one of the five most common causes of cancer death with 782,000 deaths [1]. Chemotherapy, radiotherapy, and liver transplantation are the treatment options available for treating liver cancer [2]; however, no perfect treatment has been developed thus far. Therefore, an alternative therapy for treating liver cancer is urgently needed. Epidemiological evidence and several studies show that a diet that is rich in natural bioactive compounds is associated with lower risk of human diseases [3]. Hence, many studies have strived to search for natural chemopreventive agents to reduce the mortality caused by liver cancer.

Djulis (*Chenopodium formosanum*) is a native cereal plant in Taiwan. Djulis has been proved to show marked biological activities such as antioxidant potential [4], antihypertension [5], anti-adipogenesis [6], hepatoprotective activity [4,7], and anti-cutaneous aging effect [8]. Moreover, djulis contains valuable nutrients and bioactive compounds [5], which contribute to its biological activities [5]. Therefore, djulis is widely used as a functional food, and its functional properties are receiving increasing attention. A husk is the outer shell or coating of a seed. Traditionally, the husk is discarded as an agricultural waste byproduct, with little value. However, the residues of plants have been proven to exhibit natural biological properties [9]. Thus, recent studies have focused on the potential of plant residues to serve as phytochemicals or chemopreventive agents against various diseases. Djulis husk, an example of this type of agricultural waste, is removed during the husking process with a husking separator. Djulis husk contains higher levels in crude ash, minerals, total polyphenols, and flavonoids compared to whole djulis [10]. In addition, the antihyperglycemic activity of djulis husk has been investigated [11]. Beside these studies, to the best of our knowledge, other biological activities of djulis husk have been less regarded. Given that djulis husk has biological effects and contains bioactive compounds, it is also possible that anticancer activity may be shown by djulis husk. However, no reports so far have shown the effectiveness of djulis husk in inhibiting hepatoma cell growth. Therefore, the aim of this study is to explore whether the antiproliferative effect of djulis husk and its bioactive compounds affect human HepG2 cells and the mechanism of action.

## 2. Materials and Methods

### 2.1. Materials

HepG2 cells (BCRC60025) were obtained from the Bioresource Collection and Research Center (BCRC, Food Industry Research and Development Institute, Hsinchu, Taiwan). HepG2 cells were cultured in minimum essential medium (MEM) containing 10% fetal bovine serum, 2 mM glutamine, 100 units/mL of penicillin, and maintained in humidified 5% CO_2_/95% air at 37 °C. The husk of djulis (*Chenopodiun formosanum*) purchased from Colaidea Co., Ltd., Pingtung, Taiwan. All chemicals were of analytical reagent grade.

### 2.2. Sample Preparation

The husk of djulis (10 g) was extracted with 50% ethanol (100 mL), stirred for 8 h and stood over night at room temperature. The extract was filtered, and the residue was re-extracted under the same conditions. The filtrates were combined and concentrated under reduced pressure and freeze-drying. The ethanolic extract of djulis husk, abbreviated as EEDH, was stored at −20 °C until used.

### 2.3. Cell Viability Assay

The tetrazolium dye colorimetric test (MTT assay) was done to determine cell survival [4]. The dye MTT (3-[4,5-dimethylthiazol-2-yl]-2,5-diphenyltetrazolium bromide) was purchased from PanReac AppliChem (Darmstadt, Germany). The culture media containing EEDH (50–500 µg/mL), kaempferol or quercetin (25–150 µM) were added to each of the 96 wells, and the cells were incubated in humidified 5% CO_2_/95% air at 37 °C for 72 h, with untreated cells served as control. After a period of incubation, 50 μL of a 0.1% MTT solution was added, and the cells were incubated for a further 1 h. Subsequently, the reaction was terminated and the plates were incubated for 30 min to solubilize the formazan dye by addition of dimethyl sulphoxide. The optical density of each well was determined at 550 nm using an ELISA reader (Molecular Devices, VMax, Visalia, CA, USA).

### 2.4. Lactate Dehydrogenase (LDH) Assay

The cell leakage rate was determined by LDH Cytotoxicity Colorimetric Assay Kit II (BioVision Inc., Milpitas, CA, USA). HepG2 cells were incubated at a density of 5 × 10^4^ cells/mL in a 24-well plate for 24 h. The new culture media (1 mL) containing test samples, as described above, were added to each of the 24 wells, and the cells were incubated for 72 h. At the end of the incubation, 10 μL of media from each of the 24 wells was transferred to a 96-well plate, and 100 μL of LDH reagent was added and incubated for 30 min in dark at room temperature. The absorbance was read at 450 nm using ELISA reader (Molecular Devices). The LDH leakage was estimated from the ratio between the LDH activity in the medium and that of the whole cell content [12].

### 2.5. Measurement of Mitochondrial Membrane Potential (ΔΨm)

HepG2 cells were seeded at a density of 2 × 10^5^ cells/mL in a 6-well plate for 24 h. The new culture media (1 mL) containing test samples were added to each of the 6 wells, and the cells were incubated for 24 h. After incubation, cells were incubated with JC-1, at 37 °C for 30 min in a humidified atmosphere containing 5% CO_2_. Cells were collected and washed with 200 μL of PBS in twice, subsequently, treated with 100 µL Trypsin-EDTA for 3 min and neutralized by 500 μL culture media. After the cells were centrifuged (75.53× *g*, 10 min), they were resuspended in 100 μL of PBS and analyzed by a fluorophotometer (FLx 800, BioTek, Winooski, VT, USA) with an excitation wavelength of 530 nm and an emission wavelength of 590 nm for fluorescence measurement.

### 2.6. Measurement of Caspase-3 Activity

Caspase-3 activity was detected by using the Caspase-3/CPP32 Colorimetric Assay Kit (BioVision, Palo Alto, CA, USA) according to the manufacturer’s instructions. HepG2 cells (3 × 10^6^ cells/mL) were plated in a 10 cm dish for 24 h. The new culture media (2 mL) containing test samples were added to each dish, and the cells were incubated for 48 h. Cells were collected and washed with PBS as same as describe above. Briefly, 150 μg of protein, in a total volume of 50 μL, was added to 50 μL of reaction buffer and 5 μL of DEVD-*p*NA substrate (200 μm final concentrations). After incubation (1–2 h, 37 °C), DEVD-*p*NA cleavage was monitored by detecting enzyme-catalyzed release of *p*NA at 405 nm using a microplate reader (Molecular Devices).

### 2.7. Evaluation of Reactive Oxygen Species (ROS) in HepG2 Cells

The generation of intracellular reactive oxygen species (ROS) was determined by 2’7’-dichlorofluorescin diacetate DCFH-DA [13]. HepG2 cells (2 × 10^5^ cells/mL) were pretreated with DCFH-DA (50 μM) for 30 min, and samples were added to the medium. After incubation at 37 °C for 16 h, ROS produced from intracellular stress was detected using a Bio-Tek FLx 800 microplate fluorescence reader with excitation and emission wavelengths of 485 and 528 nm, respectively.

### 2.8. Apoptosis Assay by Flow Cytometry

The apoptosis of cells induced by the extract was assayed by treatment with Annexin V and propidium iodide (PI) double labeling according to the manufacturer’s instruction (BioVision Annexin V-FITC Apoptosis Detection Kit). HepG2 cells (2 × 10^6^ cells/mL) were incubated in a 6 cm dish for 72 h in the presence or absence of the test samples. Cells were centrifuged to remove the medium, washed with PBS, and stained with Annexin V and PI in binding buffer. The stained cells were determined using FACScan flow cytometry (, Beckman Coulter, Indianapolis, IN, USA).

### 2.9. DAPI Nucleic Acid Staining

After HepG2 cells (5 × 10^4^ cells/mL) were cultured with test samples in 24-well plates for 72 h; the cells were harvested, washed with PBS, and fixed with 4% paraformaldehyde in PBS for 30 min. Fixed cells were washed with PBS and stained with 40,60-diamidino-2-phenylindole (DAPI). Evaluation was performed by a fluorescence microscope using a 350 nm excitation and a 461 nm filter for detection.

### 2.10. Cell Cycle Analysis

Cell cycle was assayed by flow cytometry. The cells were seeded at a density of 2 × 10^6^ cells/mL in a 6 cm dish and treated with the test samples for 72 h. The cells were washed twice with PBS and fixed in 70% ice-cold ethanol overnight. The sample was concentrated by removing ethanol and treating with 800 μL PBS, adding 100 μL RNase (0.2 mg/mL), incubating at 37 °C for 2 h, then adding 100 μL propidium iodide and incubating at room temperature for 15 min in dark. The cell cycle distribution was analyzed by flow cytometry [14].

### 2.11. Western Blot Analysis

The expression of Bax, Bcl-2, and poly(ADP-ribose)polymerase (PARP) related to apoptosis was measured by Western blot. In brief, after treatment with the test samples for a period of time, cells were harvested and lysed in the ice-cold lysis buffer and kept on ice for 30 min. After centrifugation at 75.53 g for 10 min at 4 °C, the supernatants were collected and the protein contents of lysates were determined by the Bicinchoninic acid (BCA) method (Pierce, Rockfold, IL, USA). Protein sample were separated using SDS-PAGE and electrically transferred to a NC membrane (Pall, New York, NY, USA). Then, the membranes were blocked with TBST (50 mM Tris-HCl, pH 7.4, 0.15 M NaCl, 0.1% Tween-20) containing 5% BSA (Sigma, St. Louis, Danvers, MA, USA) at 4 °C over night. After washing with Tris Buffer Saline Tween-20 (TBST) buffer three times, the membranes were incubated with primary antibodies against Bax (#2772, Cell Signaling Technology, Danvers, MA, USA), Bcl-2 (#3498, Cell Signaling Technology, Danvers, MA, USA), PARP (#9542, Cell Signaling Technology, Danvers, MA, USA), and β-actin (#4970, Cell Signaling Technology, Danvers, MA, USA) protein diluted at 1:1000 at room temperature for 2 h. After washing with TBST buffer three times, the membranes were incubated with goat anti-rabbit IgG labeled with horse radish peroxidase (Santa Cruz, Dallas, TX, USA) diluted at 1:5000 at room temperature for 2 h. After washing with TBST buffer three times, blots were developed using an Enhanced Chemiluminescence (ECL) plus kit (Amersham Bioscience, Aylesbury, UK), exposed to Kodak autoradiographic films and quantified using Image J software (version 1.52v, National Institutes of Health, Bethesda, MD, USA). The bands were visualized by densitometry and analyzed.

### 2.12. The HPLC-Tandem MS Analysis of EEDH

The composition of EEDH was analyzed by using the HPLC/electrospray ionization mass spectrometer (LC-ESI-MS) as described in [15] with slight modification. In brief, ten μL of prepared sample was separated with the Waters HSS T3 (2.1 × 150 mm, 1.8 µm, Waters Corp., Milford, MA, USA) analysis column fitted with a guard column (2.1 mm × 2.0 mm, sub-2µm, Security-Guard Ultra C18, Phenomenex, Inc., Torrance, CA, USA) with the Agilent 1200 HPLC system. The gradient elution was composed of two solvents: Solvent A (water containing 0.1% formic acid) and Solvent B (acetonitrile containing 0.1% formic acid) in a flow rate of 0.2 mL/min at the column temperature of 35 °C. The binary gradient elution was conducted as follows: 0–3 min (2% B), 3–6 min (2–10% B), 6–25 min (10–45% B), 25–30 min (45–95% B), 30–40 min (95% B isocratic elution), and 40–45 min (95–2% B). The absorption spectra of eluted compounds were scanned within 210 to 600 nm using the in-line diode-array detector (DAD). The separated compounds were further identified with the Agilent 6420 triple quadruple mass spectrometer. The positive and negative electron spray ionization modes were applied in the analysis in a potential of + and −3700 V, respectively, applied to the tip of the capillary. A flow rate of 10 L/min of nitrogen and a pressure of 30 psi were used as the drying and nebulizing gas maintaining at 325 °C. The fragmentor voltage and the in-source collision induced dissociation (CID) voltage were 115 V and 15 V, respectively, in a nitrogen collision gas. The mass spectra provided by ESI-MS and ESI-MS/MS were compared with those of authentic standards when available for the identification of each separated compound.

### 2.13. Statistical Analysis

Data are expressed as mean ± SD, and ANOVA was conducted by using the SPSS software (version 12.0, SPSS Inc., Chicago, IL, USA). When a significant F ratio was obtained (*p* < 0.05), a post hoc analysis was conducted between groups by using a Duncan’s multiple range tests or a Dunnett’s test. *p*-values of less than 0.05 were considered statistically significant.

## 3. Results

### 3.1. The Effect of EEDH on HepG2 Cell Growth and Induction of Apoptosis

Figure 1 shows the response of HepG2 cells to EEDH at 50–500 µg/mL for 72 h by measuring MTT assay and lactate dehydrogenase (LDH) leakage. As shown in Figure 1A, the results showed a dose-dependent decrease in cell growth, which provides strong evidence of a potent antiproliferative effect of EEDH against HepG2 cells. Meanwhile, the results from Figure 1B showed a dose-dependent increase in LDH leakage in HepG2 cells when cells were treated with 50–500 µg/mL EEDH for 72 h. LDH is released from the cells when cells are damaged. Thus, the LDH leakage may be regarded as an indicator of cytotoxicity in apoptosis research. Clearly, the increase in LDH release induced by EEDH confirms membrane perturbation and a loss of membrane integrity, thereby significantly affecting the survival of HepG2 cells.

As shown in Figure 1C, treatment of HepG2 cells for 72 h with EEDH at 500 µg/mL resulted in 87.6% of the cells exhibiting markers of apoptosis, compared with the untreated cells (11.9%). These results indicate that the cells treated with EEDH result in apoptosis.

DAPI staining is used to detect cell apoptosis. Therefore, HepG2 cell apoptosis induced by EEDH was further verified by microscopic analysis of DAPI stained cells. Figure 1D shows that with an increased concentration of EEDH, more cells with fluorescence were observed, indicating that EEDH treatment caused significant fragmentation in the chromatin and DNA rings within the nucleus of treated cells; however, the morphology was not altered in the untreated cells. This finding reveals that EEDH promoted apoptosis of HepG2 cells.

The effect of EEDH treatment on apoptosis was measured using the flow cytometry method (Figure 1E). After incubation with EEDH for 72 h, EEDH induced Sub-G0 phase cell arrest in a dose-dependent manner. The percentage of cells in the Sub-G0 phase was significantly increased to 14.1%, 28.1%, and 46.3% at concentrations of 50, 250, and 500 µg/mL, respectively, compared with 0.45% in the untreated cells. Meanwhile, the percentage of cells in the G1 phase reduced to 63.6%, 56.4%, and 41.9%, compared with 83.8% in the control. These observations suggest that the HepG2 cells were arrested in the Sub-G0 phase after EEDH treatment.

### 3.2. The Effect of EEDH on Apoptosis in Cells

To analyze the effect of EEDH on mitochondria, the effect of EEDH on the mitochondrial membrane potential in HepG2 cells was investigated. As shown in Figure 2A, when the cells were treated with EEDH for 24 h, the mitochondrial membrane potential of cells decreased to 94.3%, 55.1%, and 35.8% for 50, 250, and 500 µg/mL, respectively, compared to the control (100%), indicating that EEDH caused mitochondrial damage.

Since the examination through DAPI assay and annexin V/PI staining revealed typical features of apoptosis, the extent of apoptotic induction was further investigated. The expression of Bax and Bcl-2 was measured by Western blot analysis in HepG2 cells treated with EEDH for 24 h. Figure 2B shows a significantly elevated ratio of Bax/Bcl-2 in cells treated with EEDH. EEDH at 50, 250, and 500 µg/mL increased the Bax/Bcl-2 ratio by 1.02-, 1.17-, and 1.42-fold, respectively. No significant difference in the ratio of Bax/Bcl-2 was found with 50 µg/mL of EEDH and in the untreated cells. This result reveals that EEDH induces apoptosis in HepG2 cells through alternation of the Bax/Bcl-2 ratio.

Caspase-3 is a key executor in the apoptotic mode of cell death. The effect of EEDH on caspase-3 activity was determined. As shown in Figure 2C, incubation of HepG2 cells with EEDH at 250 and 500 µg/mL caused 29.3% and 84.3% increases in caspase-3 activity, respectively, compared with the control cells, indicating that EEDH significantly induced caspase-3 activity in HepG2 cells. In addition, activation of caspase-3 leads to the cleavage of PARP. Although PARP is not essential for cell death, the cleavage of PARP is regarded as a hallmark of apoptosis. Therefore, the cleavage of PARP in cells treated with EEDH was examined. As expected, PARP was also cleaved after the cells were treated with EEDH for 56 h. (Figure 2D). EEDH induces PARP cleavage in HepG2 cells in a dose-dependent manner. These results suggested that EEDH treatment induces caspase-dependent apoptosis in HepG2 cells.

### 3.3. ROS Generation Contributes to Apoptosis

To further understand whether EEDH induces intracellular ROS generation in HepG2 cells, the DCFH-DA model was used. As expected, a dose-dependent increase in the generation of ROS was observed when the cells were exposed to EEDH for 16 h (Figure 2E). EEDH at 250 and 500 µg/mL significantly induced ROS generation.

### 3.4. Bioactive Compounds Presented in EEDH

Attention has been given to the chemopreventive effect of bioactive compounds in natural sources. Therefore, the bioactive compounds in EEDH were analyzed and identified using HPLC-DAD and HPLC/ESI-MS/MS. Figure 3 shows the HPLC-MS total ion and HPLC-DAD chromatograms for EEDH. The bioactive compounds of EEDH from the HPLC-DAD and HPLC/ESI-MS/MS analysis with respect to retention time, λ-max in the ultraviolet region, molecular ion, and main fragment ions in MS/MS are given in Figure 3 and Table 1. Eighteen compounds were identified and quantified by comparing retention times, commercial standards, and literature data. Of these eighteen compounds, fourteen compounds, including betaine (1), betaxanthins derivatives (2), 3,4-dihydroxy-L-phenylalanine (3), amaranthin (5), isoamaranthin (6), betanin (7), isodopaxanthin (8), isobetanin (9), dihydroxybenzoic acid-*O*-dipentoside (10), quercetin-3-trisaccharide (12), amaranthin isomer (13), rutin (14), 20-hydroxyecdysone (15), and kaempferol 3-*O*-β-rutinoside (16) are identified (Table 1). Of these, betaine (2902.7 µg/g), rutin (909.9 µg/g), quercetin-3-trisaccharide (680.3 µg/g), and kaempferol 3-*O*-β-rutinoside (625.5 µg/g) are the four most abundant compounds in EEDH quantified by using the peak of HPLC-DAD analysis. Rutin (quercetin-3-*O*-rutinose), which is composed of one molecule of quercetin as aglycone and rutinose, is ubiquitously present in plants. In addition, glycosides, such as quercetin-3-*O*-rutinose and kaempferol 3-*O*-β-rutinoside, can be metabolized into aglycones by the colon microflora and transferred to tissues through systemic circulation [16]. Many studies have noted that the aglycone has a greater biological effect than the glycoside flavonoid [17]. The contents of betaine were the highest of the compounds present in EEDH, however, phenolic compounds are not only well-known for their biological effect, but also the contents of rutin, quercetin-3-*O*-rutinose and kaempferol 3-*O*-β-rutinoside significantly exist in EEDH. In addition, rutin is converted into quercetin, which is readily absorbed by the small intestine [17]. Therefore, quercetin and kaempferol were selected as reference compounds to determine the inhibitory effect against HepG2 cell growth and to further elucidate the mechanism of action for anti-proliferation of EEDH. The effects of quercetin and kaempferol at 25–150 µM on HepG2 cell survival and apoptosis are shown in Figure 4. As expected, quercetin and kaempferol in the range of 75–150 µM significantly decreased HepG2 cell growth (Figure 4A) and increased LDH leakage (Figure 4B). The data in Figure 4C show that quercetin and kaempferol increased the levels of apoptotic bodies. Moreover, the treatment of quercetin and kaempferol at 75 and 150 µM on HepG2 cells led to a significant decrease in mitochondrial membrane potential (Figure 4D), increase in Bax/Bcl-2 ratio (Figure 4E), and increase in caspase-3 activity (Figure 4F) compared to the untreated cells. Apparently, quercetin and kaempferol in the tested concentrations suppressed HepG2 cell growth and induced the cell apoptosis, which may in part account for the antiproliferative effect of EEDH on HepG2 cells.

## 4. Discussion

Recently, many chemopreventive agents from natural resources have been investigated and identified as effective inhibitors to various cancer cells. In this study, the anticancer activity of EEDH was evaluated against hepatoma carcinoma cell line HepG2. MTT assay has been widely used to test cell viability. In addition, LDH is a stable cytoplasmic enzyme present in all cells. The release of intracellular enzymes can be regarded the cells as damaged [18]. As shown in Figure 1A, EEDH significantly inhibited HepG2 cell growth proportional to its concentration. Thus, EEDH showed cytotoxicity in human hepatoma carcinoma cell HepG2 cells. To further verify the findings from HepG2 cells, LDH leakage into the culture medium was investigated. As expected, EEDH also increases the release of LDH in HepG2 cells (Figure 1B). These results reveal that EEDH has a cytotoxic effect on HepG2 cell survival, possibly due to disruption of the cell membrane. The mechanism of action of EEDH was further investigated by using annexin V/PI staining, which detects the externalization of phosphatidylserine (PS). According to the results from Figure 1C, treatment of HepG2 cells with EEDH for 72 h induced cell-surface annexin V binding, thus showing the ability of EEDH to induce apoptosis of HepG2 cells.

Classical apoptosis is characterized by key morphological features, such as membrane blebbing, chromatin condensation, and the generation of apoptotic bodies [19]. Thus, microscopic imaging of EEDH treated HepG2 cells was used to confirm the results of the anti-proliferative action of EEDH. DAPI is a highly specific stain that preferentially binds to TA regions of the DNA molecule. Therefore, DAPI assay is widely used to study the changes in DNA and analyze DNA content during apoptosis. As shown in Figure 1D, the percentage of cells stained by DAPI was significantly different among EEDH-treated and untreated cells. Apparently, chromatin condensation-related changes in EEDH-treated cells were observed. This DAPI staining result indicates that morphological changes in the nuclei of HepG2 cells were induced by EEDH. In other words, EEDH induced DNA damage in the HepG2 cells.

Cell cycle arrest is often found during programmed cell death. In addition, apoptosis occurs simultaneously with cell cycle arrest by interfering with the particular cyclin involved [20]. Therefore, the induction of cell cycle arrest is an effective method to prevent uncontrolled cell proliferation [21]. The data shown in Figure 1E clearly indicate that EEDH blocks the Sub-G0 phase in a dose-dependent manner while simultaneously suppressing cell growth (Figure 1A). Apparently EEDH inhibits the HepG2 cell growth, due in part to via the induction of cell cycle arrest in the Sub-G0 phase [22]. A probable explanation may be that EEDH impacts the cyclin involved, thus regulating the Sub-G0 phase [20].

Disruption of ΔΨm is one of the earliest indicators of induction of intracellular damage [23]. To determine whether EEDH promotes or inhibits apoptosis, the ΔΨm was examined after the cells were treated with EEDH. As shown in Figure 2A, the ΔΨm in HepG2 cells decreased in a concentration-dependent manner. The results from Figure 1A and Figure 2A,E showed that the loss of ΔΨm paralleled with the activation of caspase-3 inhibition of cell growth. Clearly, EEDH promotes ΔΨm collapse and subsequently coordinate caspase cascade activation, thereby leading to cell apoptosis [24].

Bcl-2 family members and caspases are key downstream effectors of apoptosis [19,25]. Among the Bcl-2 family, both Bax and Bcl-2 are major regulators of the intrinsic mitochondria-mediated pathway to apoptosis. Bax can act on the mitochondria to induce mitochondrial permeability transition, resulting in the release of various components, including cytochrome c [26]. Meanwhile, Bcl-2 normally suppresses apoptosis by inhibiting the release of mitochondrial proteins and phosphatidyl serine (PS) exposure [27]. That is to say, the ratio of Bax/Bcl-2 is believed to play an important role in determining apoptosis [28]. In addition, the results demonstrated that Bax was translocated to mitochondria, leading to cytochrome c release from mitochondria into the cytosol in response to EEDH [26], subsequently, enhances activation of caspase-9, and then binds to Apaf-1, thereby leading to the activation of caspase-3 and PARP cleavage [29]. Although, the protein expression of caspase-9 and Apaf-1 were not investigated in the current study, EEDH induced a remarkable expression of caspase-3 activity and enhancement of PARP cleavage, suggesting that apoptosis might be initiated through an internal mitochondrial pathway [27]. In other words, the over-expressions of caspase-3, PARP cleavage, and pro-apoptotic Bax, accompanied by under-expressed anti-apoptotic Bcl-2, indicated that EEDH induced HepG2 cell apoptosis via a mitochondrial pathway [30].

Accumulating studies have revealed that cancer cells exhibit altered redox status compared with normal cells, including increased ROS generation and copper to maintain their malignant phenotypes and thereby are more sensitive and vulnerable to further ROS production [31]. In addition, many studies noted that natural compounds such as ellagic acid, epigallocatechin, resveratrol, and anthocyanins demonstrate a pro-apoptotic effect, which relates in particular to the increased intracellular accumulation of ROS [32]. That is to say, ROS plays an important role in the induction of apoptosis. As shown in Figure 2E, there was a statistically significant increase in ROS generation when EEDH was treated with HepG2 cells. Indeed, for treatment with EEDH at 500 µg/mL, there was approximately 0.34 times higher ROS generation at the intracellular level when compared to the control. This significant increase in ROS generation was accompanied by inhibition of cell growth, cell cycle arrest, loss of MMP, increase in Bax/Bcl-2 ratio, up-regulation of caspase-3 activity, and PARP cleavage. In other words, the ROS generation induced by EEDH may contribute to apoptosis of HepG2 cells.

Numerous epidemiological evidence has suggested that the frequency of natural bioactive compound consumption relates inversely to incidence of heart disease and associated mortality, total cancer, and all-cause mortality [33]. According to Figure 3 and Table 1, EEDH comprises polyphenolics and other nonphenolic compounds, which have been widely used in the functional food industry. In an attempt to elucidate which bioactive compounds identified in EEDH induced the apoptosis of HepG2 cells, the effects of the bioactive compounds on antiproliferative action of HepG2 cells were examined. As expected, quercetin and kaempferol showed antiproliferative activity in HepG2 cells. Therefore, EEDH demonstrated marked the suppression of HepG2 cell growth, probably partly due to the quercetin and kaempferol present in EEDH. In addition, quercetin and kaempferol significantly induced the apoptosis of HepG2 cells at the utilized concentrations, as compared to a control group, as demonstrated by Annexin V/PI staining. Moreover, quercetin and kaempferol significantly increased the apoptotic bodies, reduced MMP, enhanced the ratio of Bax/Bcl-2, activated caspase-3 activity, and increased PARP cleavage levels. Apparently, quercetin and kaempferol inhibit HepG2 cell growth through apoptosis by activating intrinsic pathways. In addition, the results showed that EEDH inhibited HepG2 cells growth, in parallel with the inhibition of HepG2 cell growth by quercetin and kaempferol, indicating both bioactive compounds may account for suppression of HepG2 cell growth. It is worth noting that compounds such as betaine, betanin, isobetanin, rutin, 20-hydroxyecdysone, and other uncharacterized compounds are present in EEDH, along with quercetin derivative and kaempferol derivative. These compounds have been shown to induce biological activity. For example, betanin is permitted for use in food and pharmaceutical products as a natural red colorant. In addition, betanin may demonstrate biological activities [5,34]. The compound 20-hydroxecdysone is one of the most common plant-derived ecdysteroides. Ecdysteroides exhibit biological effects including lowering cholesterol levels and blood glucose and hepatoprotective activity [35]. Rutin is an anticancerous agent against cervical cancer and colon cancer [34,36]. Betaine is distributed widely in many animals, plants, and microorganisms. In addition, the growing body of evidence shows that betaine is an important nutrient for the prevention of chronic disease [37]. Although the role of betaine responsible for its antiproliferative effect was not investigated in this study, it is suggested that betaine could potentially induce apoptosis on HepG2 cell. However, this speculation requires further study. Apart from betaine, there were thirteen identified compounds and another four uncharacterized compounds observed in EEDH. In other words, these compounds and other uncharacterized compounds may contribute to a direct and also synergistic effect of a combination of bioactive compounds present in EEDH [6].

## 5. Conclusions

In summary, these results demonstrated for the first time that EEDH exerts significant anti-proliferative activity and induces apoptosis in HepG2 cells through a mitochondrial pathway. In addition, EEDH-induced apoptosis in HepG2 cells might be attributable to its bioactive compounds such as quercetin, kaempferol, and other compounds that are present in EEDH. Therefore, these results provide new insight into the chemoprevention of EEDH, which may prove to be a potent new anticancer compound for treatment of hepatoma cells. In addition, how the effectiveness of EEDH is based on the concentrations used. However, before EEDH is used as a direct chemopreventive agent in clinics, further in vivo experiments and assessments need to be conducted.

## Figures and Tables

**Figure 1 foods-09-01514-f001:**
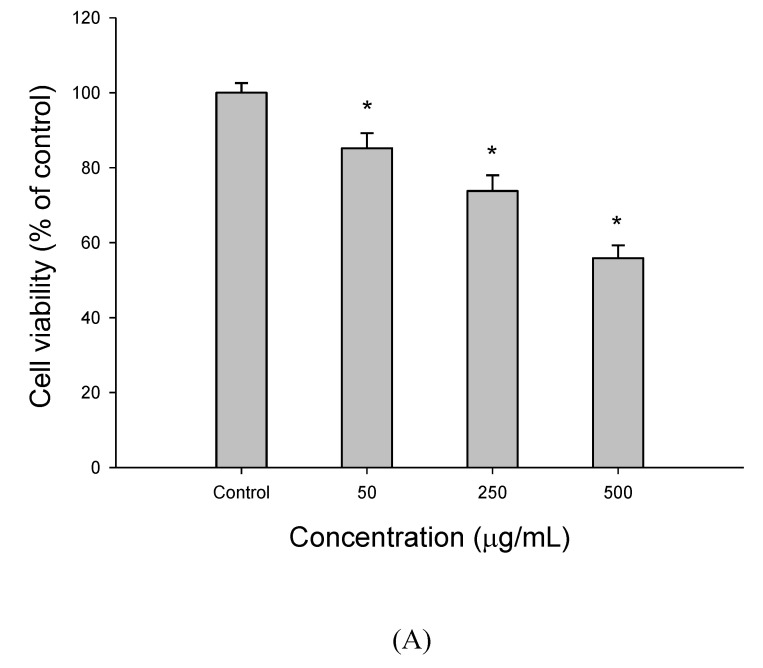
The cell viability and apoptotic effect of ethanolic extracts of djulis husk (EEDH) on HepG2 cells. (**A**) Effect of EEDH on cell viability. After EEDH-treated cells for different incubation time, cell viability was determined by MTT assay. (**B**) Effect of EEDH treatment on cell disruption was determined by lactate dehydrogenase (LDH) leakage assay. The results are expressed as percent of LDH activity in the culture medium relative to the total enzyme activity. (**C**) Effect of EEDH on apoptosis in HepG2 cells. The cells were harvested and stained with Annexin V-FITC and propidium iodide (PI) and determined by flow cytometry. (**D**) Chromatin condensation was determined by 40,60-diamidino-2-phenylindole (DAPI) staining after EEDH treatment. The DAPI-stained cells were evaluated using fluorescence microscopy (200 ×). The arrow indicated apoptotic cells. (**E**) Effect of EEDH on the cell cycle of HepG2 cells. The cells were incubated with EEDH for 72 h (**A**–**E**). The data are expressed as the mean ± SD (*n* = 3). * indicated significant difference from the control (*p* < 0.05).

**Figure 2 foods-09-01514-f002:**
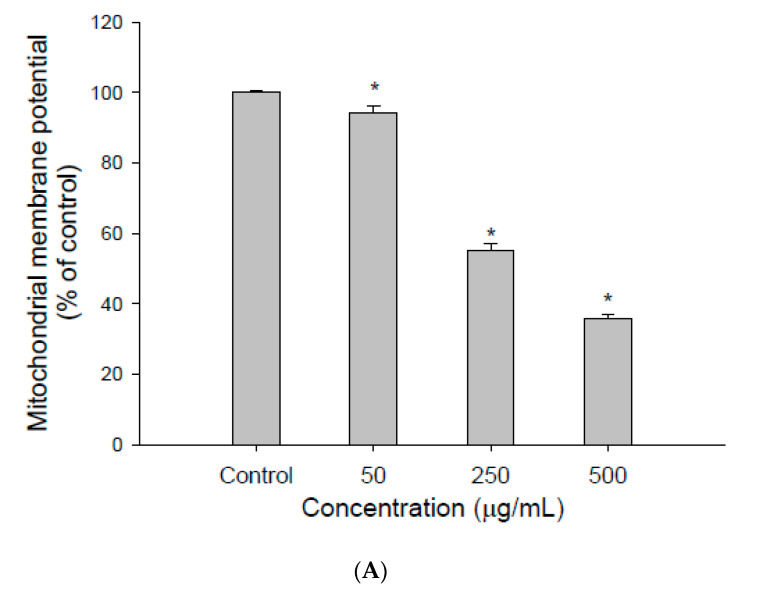
The effect of different concentrations of ethanolic extracts of djulis husk (EEDH) on HepG2 cells apoptosis. (**A**) Effect of EEDH on mitochondrial membrane potential in HepG2 cells. (**B**) Effect of EEDH on Bax/Bcl-2 ratio in HepG2 cells. (**C**) Effect of EEDH on caspase-3 activity in HepG2 cells. (**D**) Effect of EEDH on PARP cleavage in HepG2 cells. (**E**) Effect of EEDH on reactive oxygen species (ROS). The cells were incubated with EEDH for 24 h (**A**), 24 h (**B**), 48 h (**C**), 56 h (**D**), and 16 h (**E**), respectively. The data are expressed as the mean ± SD (*n* = 3). * indicated significant difference from the control (*p* < 0.05).

**Figure 3 foods-09-01514-f003:**
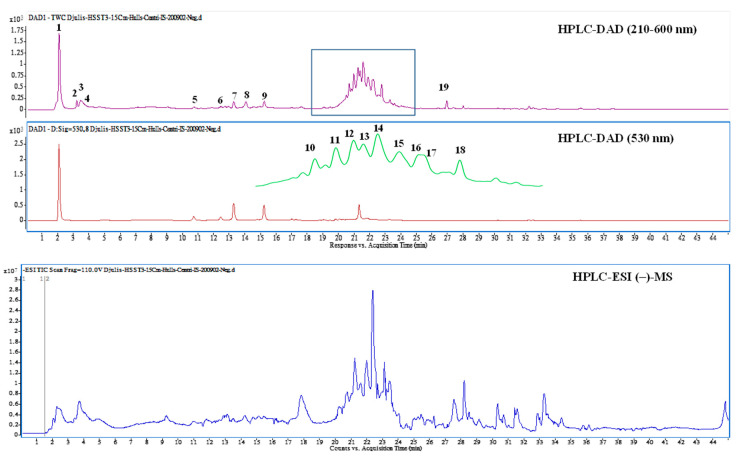
High-performance liquid chromatograms detected at a full UV-Vis spectrum of 210–600 nm (top) and UV 530 nm (middle), and the total ion chromatogram of negative ionization mass spectrometry (bottom) from ethanolic extracts of djulis husk.

**Figure 4 foods-09-01514-f004:**
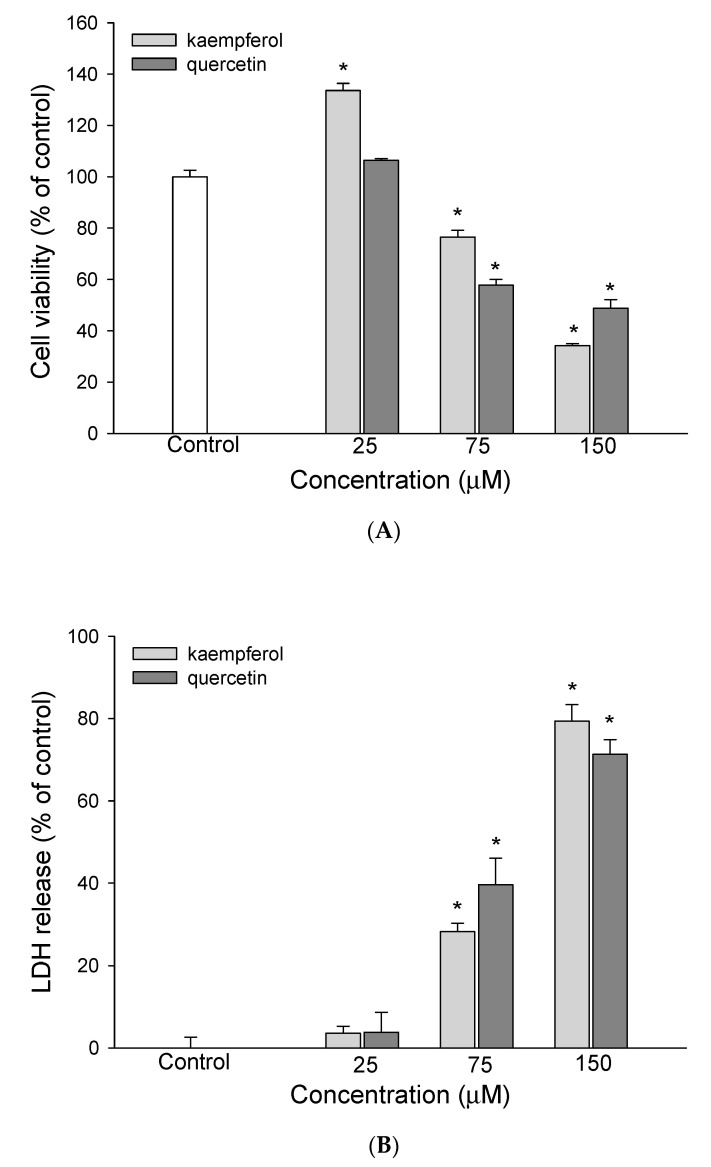
The effect of kaempferol and quercetin on cell viability and apoptotic effect of HepG2 cells. (**A**) Effect of kaempferol or quercetin on cell viability was determined by MTT assay. (**B**) Effect of kaempferol or quercetin on cell disruption was determined by lactate dehydrogenase (LDH) leakage assay. The results are expressed as percent of LDH activity in the culture medium relative to the total enzyme activity. (**C**) Chromatin condensation was determined by DAPI staining after kaempferol or quercetin treatment. The DAPI-stained cells were evaluated using fluorescence microscopy (200 ×). The arrow indicated apoptotic cells. (**D**) Effect of kaempferol or quercetin on mitochondrial membrane potential in HepG2 cells. (**E**) Effect of kaempferol or quercetin on Bax/Bcl-2 ratio in HepG2 cells. (**F**) Effect of kaempferol or quercetin on caspase-3 activity in HepG2 cells. The cells were incubated with kaempferol or quercetin for 72 h (**A**), 72 h (**B**), 72 h (**C**), 24 h (**D**), 24 h (**E**), and 48 h (**F**), respectively. The data are expressed as the mean ± SD (*n* = 3). * indicated significant difference from the control (*p* < 0.05).

**Table 1 foods-09-01514-t001:** Retention time, UV-Vis, and mass spectral characteristics of ethanolic extracts of djulis husk (EEDH).

Peak No.	RT (min)	Compound Name	λ_max_ (nm)	[M + H]^+^/[M − H]^−^, *m/z*	MS/MS *m/z* ^c^	Content (μg/g)
1	2.04	Betaine *	226, 278, 532	**118**^b^/	58, 59	2902.7 ± 118.3
2	3.17	Betaxanthins derivatives *	222, 264, 482	**280**/	216, 84, 198, 97	220.1 ± 38.5
3	3.34	3,4-Dihydroxy-L-phenylalanine	224, 280	**198**/	152, 135, 107, 139	392.6 ± 35.8
4	3.61	Unknown	224, 280	**360**/	152, 164, 296, 139	174.0 ± 20.9
5	10.67	Amaranthin	268, 536	**727**/	389	72.4 ± 6.5
6	12.39	Isoamaranthin	264, 530	**727**/	389	62.4 ± 18.5
7	13.23	Betanin ^a^	260, 290sh, 538	**551**/	389	290.0 ± 21.8
8	14.19	Isodopaxanthin	260, 472	**391**/389	255, 150, 345, 347	314.1 ± 24.2
9	15.24	Isobetanin	268, 290sh, 532	**551**/	389	220.9 ± 45.9
10	20.68	Dihydroxybenzoic acid-*O*-dipentoside	224, 260, 332	**417**	152, 108	439.8 ± 101.5
11	20.97	Unknown	228, 254, 344	**449**/	337, 199, 127, 215	595.5 ± 71.6
12	21.24	Quercetin-3-trisaccharide	228, 254, 322	**743**/	303	680.3 ± 45.3
13	21.39	Amaranthin isomer *	224, 328, 528	**727**/	389	292.3 ± 64.7
14	21.59	Rutin ^a^	254, 352	**611**/609	301	909.9 ± 102.4
15	21.93	20-Hydroxyecdysone	246, 316, 422	**481**	165, 371, 301, 173	505.5 ± 25.5
16	22.01	Kaempferol 3-*O*-β-rutinoside	228, 266, 316	**595**/593	287	625.5 ± 267.2
17	22.15	Unknown	228, 282, 350	**291**/289	159, 130, 185, 227	326.3 ± 28.3
18	22.77	Unknown	224, 312, 410	**677**/675	319, 599, 643, 557	448.7 ± 46.5
19	26.93	Internal standard	220, 272, 312	**255**/	151, 131, 103, 209	250

^a^ Compound identification by comparison with authentic standards. ^b^ Values in bold indicate the molecular ion for MS/MS fragmentation. ^c^ MS/MS fragment ions are shown with decreasing order according to their signal intensity. * Tentatively identified. Internal standard: 7-methoxyflavanone.

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
