# Peer review of "Antiproliferative Effect and Mediation of Apoptosis in Human Hepatoma HepG2 Cells Induced by Djulis Husk and Its Bioactive Compounds"

_foods, 2020, doi:10.3390/foods9101514_

Round 1

Reviewer 1 Report

Line 23: … 50, 250 and 500

Line 25: … all EEDH treatments induced

Line 29: Quercetin and kaempferol at 25-150 mM showed (?) – and what about other compounds, they are also present in the extract. Are you sure that only Q and K are responsible for the effects?

Line 72: why 50% ethanol, and not 70-75%; do you have any data that 50% is the best solution?

Line 180: by using a Duncan’s multiple range tests or a Dunnett’s test. Please specify when did you use Duncan or Dunnett.

Section results contains some sentences with references. That is unusual approach. Typically that section is without references.

Conclusion: it must be precise statement referred to applied extract dosages. There is no doubt that lower or higher extract concentrations would act differently.

Author Response

Dear Reviewer 1,

Your comments and suggestions are greatly appreciated. We have checked and revised accordingly. I hope it can meet the journal standard and be acceptable for publication.

According to the reviewer’s suggestions, the unacceptable words and sentences in the text have been corrected and indicated in blue color. Please see the followings:

Reviewer 1 comment

Comment 1

Line 23: … 50, 250 and 500

Ans: 50, 250 and 500 are recruited at line 23.

Comment 2

Line 25: … all EEDH treatments induced

Ans:“EEDH treatment induced” have been corrected to “All EEDH treatment induced”. Please see line 25.

Comment 3:

Line 29: Quercetin and kaempferol at 25-150 mM showed (?) – and what about other compounds, they are also present in the extract. Are you sure that only Q and K are responsible for the effects?

Ans: Among eighteen compounds identified, betanin, rutin, quercetin and kaempferol were selected to measure inhibitory effect on HepG2 cell growth. The results showed that quercetin and kaempferol at 25-150 µM demonstrated significant inhibition of cell growth. Although betanin and rutin showed no significant inhibition of HepG2 cell growth, we suggested that there were twelve identified compounds and other four uncharacterized compounds observed in EEDH, along with quercetin and kaempferol. In other words, these compounds and other uncharacterized compounds may contribute to a direct and also synergistic effect of a combination of bioactive compounds present in EEDH.

Comment 4

Line 72: why 50% ethanol, and not 70-75%; do you have any data that 50% is the best solution?

Ans: Our concept is the less of solvent concentration used is better. So 50% ethanol was used in this study instead of higher concentration used.

Comment 5

Line 180: by using a Duncan’s multiple range tests or a Dunnett’s test. Please specify when did you use Duncan or Dunnett.

Ans: Duncan's multiple range test provides significance levels for the difference between any pair of means, regardless of whether a significant F resulted from an initial analysis of variance. Dunnett test is used to compare experiments with control group, which ones are different significantly from the control and which ones are not. Therefore, Duncan’s multiple range tests or a Dunnett’s test was used based on the data and results required.

Comment 6

Section results contains some sentences with references. That is unusual approach. Typically that section is without references.

Ans: References 16-23 are deleted, however, reference 24-25 are remained due to the explanation of converting of rutin to quercetin.

Ans: References 16-23 are deleted, however, reference 24-25 are remained due to the explanation of converting of rutin to quercetin.

The sentences in lines 207-210 are deleted. The sentences in lines 258-260 are deleted, and “ To further understand whether EEDH induces intracellular ROS generation in HepG2 cells, the DECF-DA model was used.” is recruited.

The sentences in lines 264-268 are deleted, and “Attention has been given to the chemopreventive effect of bioactive compounds in natural sources.” is recruited.

Refs 26-45 are changed to Refs 16-37.

Comment 7

Conclusion: it must be precise statement referred to applied extract dosages. There is no doubt that lower or higher extract concentrations would act differently.

Ans: “In addition, how the effectiveness of EEDH is based on the concentrations used” is added at line 432.

Warm regards

Pin-Der Duh

Reviewer 2 Report

The work seems to be of decent quality for “Foods ". However, I have some an important question for the author to include in the manuscript without these results of the manuscript are incomplete in my point of view.

  • The author mentioned that “The ethanolic extracts of djulis husk (EEDH) at 50-500 µg/ml induced remarkable cytotoxicity on HepG2 cells”. My concern is that there as we know most of the research are based on the purified compounds rather than crude mixture. Sometime crude extract can have side effect as well. So, my question is possible to get the purified compounds from extract which are responsible for the antiproliferative activity.
  • The authors mentioned that “The effect of kaempferol and quercetin on cell viability and apoptotic effect of HepG2 cells and effect of kaempferol or quercetin on cell viability was determined by MTT assay” Did they got these compounds purified from the extract?
  • Even though, Authors have included the Chromatograms in figure. I would suggestion authors to include the mass range on the X-axis instead to time. That will make reader to under the table 1. Time on X-axis in TIC or EIS doesn’t provide any information related to the masses in the crude extract.
  • Authors should include the TLC for the crude extract.

Author Response

Dear Reviewer 2,

Your comments and suggestions are greatly appreciated. We have checked and revised accordingly. I hope it can meet the journal standard and be acceptable for publication.

According to the reviewer’s suggestions, the unacceptable words and sentences in the text have been corrected and indicated in blue color. Please see the followings:

Reviewer 2 comment

The work seems to be of decent quality for “Foods ". However, I have some an important question for the author to include in the manuscript without these results of the manuscript are incomplete in my point of view.

Comment 1

The author mentioned that “The ethanolic extracts of djulis husk (EEDH) at 50-500 µg/ml induced remarkable cytotoxicity on HepG2 cells”. My concern is that there as we know most of the research are based on the purified compounds rather than crude mixture. Sometime crude extract can have side effect as well. So, my question is possible to get the purified compounds from extract which are responsible for the antiproliferative activity.

Ans: Djulis is a traditionally used as a native cereal. It is meaningful that djulis shows antiproliferative effect on hepatoma cells. Beside this, bioactive compounds present in djulis were identified and querceti and kaempferol, which were obtained from Sigma and, were used to determine the inhibitory effect on HepG2 cells and to further elucidate the mechanism of action for antiproliferation of djulis. According to the results, bioactive compounds such as quercetin and kaempferol and other bioactive compounds present in djulis may in part account for antiproliferative effect of EEDH.

Comment 2

The authors mentioned that “The effect of kaempferol and quercetin on cell viability and apoptotic effect of HepG2 cells and effect of kaempferol or quercetin on cell viability was determined by MTT assay” Did they got these compounds purified from the extract?

Ans: The bioactive compounds such as betanin, rutin, quercetin and kaempferol were identified using HPLC/ESI-MS-MS analysis. These compounds used to determine the antiproliferative effect was obtained from Sigma…………..  

Comment 3

Even though, Authors have included the Chromatograms in figure. I would suggestion authors to include the mass range on the X-axis instead to time. That will make reader to under the table 1. Time on X-axis in TIC or EIS doesn’t provide any information related to the masses in the crude extract.

Ans: The chromatogram from each peak shown in the Figure 1 (bottom panel) was the result from the sum of m/z 100-1000 scanning in each compound. Therefore, the separated and identified compound listed in Table 1 was arranged by the retention time (Rt) to present the identification results. That’s why the time shown in x-axis would be necessarily.

Comment 4

Authors should include the TLC for the crude extract.

Ans.: The thin layer chromatography TLC) was not applied in the study, owing to the very poor resolution than HPLC. In emerging analysis technologies in phytochemicals, the TLC is not only shown with the low separation capability, but also in the low compatibility with MS for the simultaneous separation and identification works. Therefore, HPLC/ESI-MS-MS analysis was used in this study.

Warm regards

Pin-Der Duh
